# Poisoning Federated Recommender Systems with Fake Users

## ABSTRACT

Federated recommendation is a prominent use case within federated learning, yet it remains susceptible to various attacks, from user to server-side vulnerabilities. Poisoning attacks are particularly notable among user-side attacks, as participants upload malicious model updates to deceive the global model, often intending to promote or demote specific targeted items. This study investigates strategies for executing promotion attacks in federated recommender systems.

Current poisoning attacks on federated recommender systems often rely on additional information, such as the local training data of genuine users or item popularity. However, such information is challenging for the potential attacker to obtain. Thus, there is a need to develop an attack that requires no extra information apart from item embeddings obtained from the server. In this paper, we introduce a novel fake user based poisoning attack named PoisonFRS to promote the attacker-chosen targeted item in federated recommender systems without requiring knowledge about user-item rating data, user attributes, or the aggregation rule used by the server. Extensive experiments on multiple real-world datasets demonstrate that PoisonFRS can effectively promote the attacker-chosen targeted item to a large portion of genuine users and outperform current benchmarks that rely on additional information about the system. We further observe that the model updates from both genuine and fake users are indistinguishable within the latent space.

### ACM Reference Format:
Anonymous Author(s). 2023. Poisoning Federated Recommender Systems with Fake Users. In *Proceedings of ACM Conference (Conference'17)*. ACM, New York, NY, USA, 11 pages. https://doi.org/10.1145/nnnnnnn.nnnnnnn

## 1 INTRODUCTION

Federated learning (FL) [18, 24] is gaining more and more attention in recent applications because it does not require any user's personal data on the server side. A prevailing application of FL is federated recommender systems (FedRecs) [2, 21, 22, 25, 33, 36, 41], where each participant holds its local interaction information and feature vector. In each global round, the server sends item embeddings to each user, and each user trains its local model using item embeddings and its user embedding. After that, the user sends the model update of item embeddings to the server. Therefore, the server can only access item embeddings, which is not sensitive for users.

However, recent studies have found that malicious users can alter the global model's behavior in FedRecs by uploading well-crafted model updates to poison the global model. Such manipulations can be divided into targeted poisoning attacks [29, 45, 48] and untargeted poisoning attacks [3, 8]. In targeted poisoning attacks, malicious users tend to promote or demote the targeted item, while in untargeted attacks, malicious users tend to downgrade the global model's overall performance. Since targeted attacks can bring direct interests to the attacker, e.g., promoting its own films in a film recommender system, it poses a great threat to the FedRecs. In this paper, we only focus on this type of poisoning attack.

Several targeted poisoning attacks have been proposed to manipulate the FedRecs [29, 45, 48]. However, existing attacks typically necessitate knowledge about the targeted FedRecs system, such as genuine users' local training data or the popularity distribution of items, which, in practice, is difficult for the attacker to acquire. For instance, PipAttack [48] leverages the popularity of each item to train a popularity estimator and then generates model updates so that the targeted item has a high popularity. FedRecAttack [29] needs to know genuine users' training data so that the attacker can estimate genuine user features in order to implement the attack. In the PSMU attack [45], malicious users generate some synthetic local training data that must closely mimic the distribution of genuine users' training data.

This paper proposes a novel poisoning attack called PoisonFRS to manipulate the FedRecs using fake users. In our proposed PoisonFRS attack, the attacker has no knowledge about genuine users (local training data and model updates) and the aggregation rule used by the server, and each fake user has no local training data. This is possible in some platforms, like Amazon Personalize [1]. In other scenarios, the interaction information of genuine users is visible, but the attacker often needs to crawl over the entire website, which is consumptive, and this abnormal behavior will be easily detected. As for local training data, since most fake users are newly registered and tailored for the attack, they cannot have local training data consistent with genuine users. Therefore, our attack poses significant practicability in real-world applications.

In our proposed PoisonFRS attack, the attacker carefully crafts the model updates for fake users such that the poisoned global model will promote the attacker-chosen targeted item to a large fraction of genuine users. Specifically, the attacker in our attack needs to use item features received from the server to estimate $k$ items with high popularity. After that, it constructs a targeted model based on the features of the selected items. At the end of each global round, each fake user sends a model update that drags the global model towards the target model. Such an attack only requires item embeddings available in the federated recommendation protocol. The attacker does not need to train malicious model updates using the embedding of fake users anymore and thus requires no training data.

We conducted extensive experiments on four real-world datasets. In our experiments, we compared our proposed PoisonFRS attack

with eight baseline attacks, which included five attacks in a centralized setting and three attacks designed for FedRecs. Additionally, we tested PoisonFRS on seven aggregation rules, namely FedAvg [24], coordinate-wise median [43], coordinate-wise trimmed-mean [43], Clip [17], Krum [4], FLAME [26], and HiCS [45]. Our results indicate that PoisonFRS is effective across all these aggregation rules, and significantly outperforms existing attacks. For instance, on the Yelp dataset, our PoisonFRS can promote the targeted item to over 70% genuine users while introducing only 0.05% fake users. We also investigated whether PoisonFRS could be detected by the server. We conducted a t-SNE [35] analysis of the targeted item model update and found that the model update of genuine users and fake users are indistinguishable in the latent space.

Our key contributions can be summarized as follows:

- We introduce a novel poisoning attack on FedRecs that uses fake users, requiring no prior knowledge of genuine user information or access to local training data.
- We systematically evaluate the performance of our proposed attack under various settings, and we find that PoisonFRS significantly outperforms baseline attacks.
- Extensive experiments demonstrate that our proposed PoisonFRS could promote the targeted item to a large fraction of genuine users with a small proportion of fake users, and our attack cannot be detected by the server.

## 2 RELATED WORK

### 2.1 Federated Recommender Systems

Recommender system is a technique used to provide personalized recommendations to users. Previous research on recommender systems mainly focuses on a centralized setting [11, 14, 19, 27, 30, 31, 39], where each user's feature and interaction data is collected at the central server. Such a setting poses a privacy threat to users because the server may leak sensitive data. To address this issue, federated recommender systems (FedRecs) have been proposed [2, 21, 22, 25, 33]. The basic framework of FedRecs is federated learning (FL). This learning scheme prevents the server from accessing users' local training data and thus ensures privacy.

Each user in FedRecs possesses its local training data, and the server allows users to train a global model (i.e., item embeddings) without disclosing their raw user-item rating data during the training phase. Specifically, FedRecs performs the following three steps in each global round (as shown in Figure 1):

**Step I.** The server sends the current item embeddings to each user or a subset of users.

**Step II.** Each user trains its local model using its training data and the received item embeddings. To be specific, in the $l$-th global training round, suppose the number of interacted items is $r$ and let $\mathbf{R} = \{(p_1, n_1), (p_2, n_2), \cdots, (p_r, n_r)\}$ denote the positive-negative sample pairs. The item embeddings received is denoted as $\mathbf{V} = \{\mathbf{v}_1^l, \mathbf{v}_2^l, \cdots, \mathbf{v}_m^l\}$. The local training objective for each user is defined by $L = -\sum_{i=1}^{r} \ln \sigma(\hat{y}_{p_i} - \hat{y}_{n_i})$ [28], where $\hat{y}_{p_i}$ and $\hat{y}_{n_i}$ respectively represent to which extent the user likes or hates the item $i$. So, the item embedding update is calculated as $\mathbf{g}^l = -\eta \nabla_{\mathbf{V}} L$, where $\eta$ is the learning rate. After that, each user uploads its item embedding update to the server.

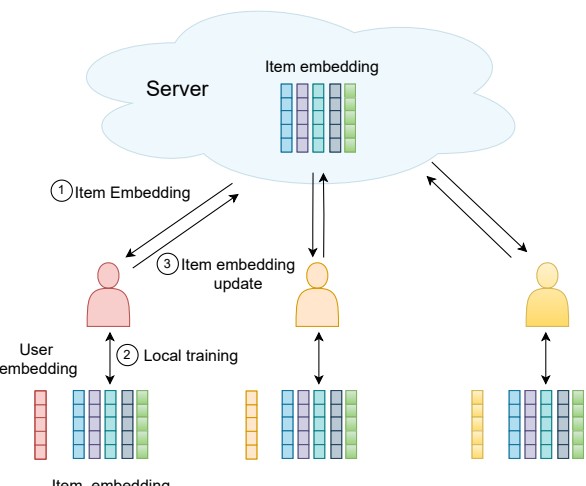

**Figure 1: Illustration of three steps in FedRecs.**

---

**Algorithm 1** Training Process of FedRecs.

---

**Input:** Number of global rounds $T$, number of items $m$, aggregation rule Agg, number of users interacted with $i$-th item $n_i$.

**Output:** Updated model for the interacted items.

1: **for** $l = 1, 2, \ldots, T$ **do**
2:     The server send item embedding $\mathbf{v}_1^l, \mathbf{v}_2^l, \cdots, \mathbf{v}_m^l$ to users.
3:     Each user trains its local model according to Algorithm 2.
4:     **for** $i = 1, 2, \ldots, m$ **do**
5:         The server receives model updates on $i$-th item $\mathbf{g}_{i,1}^l, \mathbf{g}_{i,2}^l, \cdots, \mathbf{g}_{i,n_i}^l$
6:         $\mathbf{g}_i^l \leftarrow \text{Agg}(\mathbf{g}_{i,1}^l, \mathbf{g}_{i,2}^l, \cdots, \mathbf{g}_{i,n_i}^l)$
7:         $\mathbf{v}_i^{l+1} \leftarrow \mathbf{v}_i^l + \mathbf{g}_i^l$
8:     **end for**
9: **end for**

---

**Algorithm 2** Local training for genuine users in FedRecs.

---

**Input:** Number of global rounds $T$, number of items $m$, learning rate $\eta$, number of interacted items $r$, positive-negative sample pairs $\mathbf{R} = \{(p_1, n_1), (p_2, n_2), \cdots, (p_r, n_r)\}$.

**Output:** Model updates for the interacted items

1: **for** $l = 1, 2, \ldots, T$ **do**
2:     Each user downloads item embeddings from the server.
3:     $\mathbf{V} \leftarrow \{\mathbf{v}_1^l, \mathbf{v}_2^l, \cdots, \mathbf{v}_m^l\}$
4:     $L \leftarrow -\sum_{i=1}^{r} \ln \sigma(\hat{y}_{p_i} - \hat{y}_{n_i})$
5:     $\mathbf{V}' \leftarrow \mathbf{V} - \eta \nabla_{\mathbf{V}} L$
6:     $\mathbf{g}^l \leftarrow \mathbf{V}' - \mathbf{V}$
7:     Each user uploads nonzero entries in $\mathbf{g}^l$ to the server.
8: **end for**

---

**Step III.** The server then aggregates the received item embedding update and further updates the item embeddings.

Then, the three steps are repeated until some convergence criteria are met. The complete algorithm that illustrates the whole process is given in Algorithm 1.

## 2.2 Poisoning Attacks to FedRecs

Numerous poisoning attacks [6, 9, 10, 12, 15, 16, 20, 23, 37, 47] have been proposed to manipulate recommender systems, yet the majority of these existing attacks are based on the centralized setting. Several recent studies [29, 45, 48] have demonstrated that malicious users can influence recommendation preferences in FedRecs by uploading malicious model updates to poison the system. This will make certain items promoted or demoted. However, some of them require additional information related to genuine users' training data or genuine users' training data or the distribution of items. For instance, FedRecAttack [29] requires the attacker to know genuine users' user-item rating data, and PipAttack [48] requires the attacker to know the popularities of all items. PSMU [45] requires the injected malicious users to create some synthetic local training data that resembles the distribution of genuine users' training data to achieve good attack performance. In our experiments, we demonstrate the PSMU method's limited effectiveness due to the real-world dataset's inherent sparsity. Furthermore, in PSMU, the attacker trains features of malicious users in each global round, resulting in a considerable slowdown of the attack process. Table 1 summarizes the difference between our proposed PoisonFRS attack and existing attacks.

Table 1: Knowledge required by different attacks. ○ indicates optional.

| | Genuine users' training data | Malicious users' training data | Item popularity |
|---|---|---|---|
| FedRecAttack [29] | ✓ | ✗ | ✗ |
| PipAttack [48] | ○ | ✗ | ✓ |
| PSMU [45] | ✗ | ✓(Generated) | ✗ |
| PoisonFRS | ✗ | ✗ | ✗ |

## 2.3 Byzantine-robust Aggregation Rules

To counteract attacks on FL, various Byzantine-robust aggregation rules have been proposed. These rules filter or trim malicious model updates to ensure that the aggregated model update remains relatively innocuous. Median [43] and Trimmed-mean [43] represent two typical Byzantine-robust aggregation rules. In Median, the aggregated model update is the coordinate-wise median of all model updates. In Trimmed-mean, the aggregated model update is the trimmed mean of the collected model updates. These two aggregation rules filter malicious model updates in each dimension. Alternatively, an approach involves clipping malicious model updates rather than entirely excluding them.

The aforementioned aggregation rules typically cannot completely reject malicious model updates. In Median and Trim, some dimensions of malicious model updates are inevitably included or averaged, as the aggregator cannot guarantee the exclusion of malicious updates on every dimension. In the case of clipping, malicious model updates are scaled rather than discarded. Consequently, various detect-then-drop mechanisms have been proposed. Krum [4] selects the model update closest to its neighbors for aggregation. FLAME [26] utilizes HDBSCAN [5] to implement a clustering algorithm and subsequently employs intricate processing on the selected cluster, such as clipping or adding noise.

## 3 THREAT MODEL

### 3.1 Attacker's Goal

The attacker aims to manipulate FedRecs with fake users and consequently make targeted items recommended to as many genuine users as possible. To elaborate, let $t$ denote the targeted item, $U_t$ denote the set of users who have not interacted with the targeted item $t$ yet. $V_u^{rec}$ represents the set of items recommended to user $u$, which is the set of items that has the top-$K$ predicted scores among non-interacted items of user $u$. The attacker's ultimate goal is to maximize the target hit ratio, which is defined by the following:

$$HR@K = \frac{1}{|U_t|} \sum_{u \in U_t} \mathbb{I}\left[t \in V_u^{rec}\right], \tag{1}$$

where $\mathbb{I}$ is the indicator function, $\mathbb{I}\left[t \in V_u^{rec}\right]$ is 1 if targeted item $t$ appears in $V_u^{rec}$, otherwise 0.

### 3.2 Attacker's Knowledge

In our attack model, the attacker has no knowledge about the local training data of genuine users, item distribution, and the aggregation rule used by the server. The attacker only has access to the item embedding sent by the server.

### 3.3 Attacker's Capabilities

The current predominant methods for attacking FL-related systems can be categorized into two groups: comprising genuine users and injecting fake users into the systems. However, comprising genuine users is costly, demanding significant effort from the attacker. For example, the attacker may need to employ sophisticated techniques to gain control over these genuine users and continually avoid detection. Alternatively, the attacker might need to incentivize the comprised users, essentially paying them to collaborate with the attacker.

However, injecting fake users into FL systems appears to be a more viable approach. Firstly, the attacker no longer needs to employ a series of attack methods to manipulate genuine users, as they can utilize their own devices to carry out the attack, with a single device capable of impersonating multiple fake users. Moreover, the attacker is intimately familiar with the device, significantly enhancing efficiency. Given these considerations, we conduct the attack by injecting fake users into FedRecs. Moreover, these injected fake users lack local training data and are not required to create synthetic data throughout the training procedure. These fake users could send carefully crafted item embedding update to the server.

## 4 OUR ATTACK

### 4.1 Motivation

We have identified that the reason existing attacks necessitate access to local training data is their reliance on the training of local item features to generate a malicious model update. This training procedure invariably involves a loss function that incorporates user features, thereby mandating access to local training data. To mitigate the need for fake users to have local data, an alternative approach is to abstain from training the global model altogether. Instead, fake users can pre-construct a target model, and during

each global round, these fake users compute model updates directly aimed at aligning the global model with the target model.

The primary challenge now lies in constructing this target model. To accomplish this, the attacker must discern which item feature garners the most popularity without knowledge of user features. Indeed, if the attacker can identify an item feature that scores positively with the majority of users, enhancing that feature can further promote the item. We establish the targeted item feature by aggregating popular item features. The remaining challenge is: how can one roughly identify popular items without access to users' feature vectors? Our approach is grounded in the assumption that most items are inherently unpopular, causing the average of all item features to be unpopular as well. Consequently, popular items must exhibit significant dissimilarity from the average item feature. Drawing inspiration from this insight, we select $k$ items whose features exhibit the smallest inner product with the average item feature.

In each global training round, our proposed PoisonFRS attack contains the following four steps.

- Select $k$ item embeddings that have the highest estimated popularity.
- Construct a target model with its targeted item embedding derived as the product of the averages of the $k$ item embeddings chosen above.
- Select filler items.
- Send the crafted model updates to the server in order to steer the global model toward the target model.

## 4.2 Description

This section presents the detailed design of our PoisonFRS attack.

### 4.2.1 Estimating $k$ Popular Items.
The initial task involves selecting $k$ items with the highest estimated popularity. Assuming that each user receives the global model at the $l$th global round, denoted as $\mathbf{v}_1^l, \mathbf{v}_2^l, \ldots, \mathbf{v}_m^l$, where $m$ represents the total number of items. The procedure for estimating popularity consists of the following steps: Firstly, compute the average of item features for that particular global round as $\mathbf{v}_{\text{avg}} = \frac{1}{m} \sum_{i=1}^m \mathbf{v}_i^s$, where $m$ is the number of items. Then, the fake user computes the inner product between each item feature and $\mathbf{v}_{\text{avg}}$, and we select the $k$ items that have the lowest result as the estimated most popular $k$ items.

### 4.2.2 Constructing the targeted item embedding.
In the former step, the attacker has already selected $k$ items that are estimated as popular. These items are not certainly the $k$ most popular items, but their popularity is estimated to be very high. The feature of the targeted item should be close to those $k$ items. Therefore, we can minimize their mean $\ell_2$ distance:

$$\min_{\mathbf{v}_t} \frac{1}{k} \sum_{i \in \mathbf{I}_{\text{pop}}} \|\mathbf{v}_t - \mathbf{v}_i^s\|_2^2, \tag{2}$$

where $\mathbf{v}_t$ is the feature of the targeted item, and $\mathbf{v}_i^s$ are features of selected $k$ items at the $s$-th global round, where $i = 1, 2, \cdots k$. The solution of the above optimization problem is $\mathbf{v}_t = \frac{1}{k} \sum_{i \in \mathbf{I}_{\text{pop}}} \mathbf{v}_i^s$.

However, in this way, the hit ratio of the targeted item is not much better than that of the selected $k$ items. To further improve the predicted score of the targeted item, we can multiply $\mathbf{v}_t$ by a

factor $\lambda > 1$. This is equivalent to multiplying the predicted score by $\lambda > 1$. We finally formulate the targeted item embedding in the target model as the following:

$$\mathbf{v}_t' = \lambda \mathbf{v}_t = \frac{\lambda}{k} \sum_{i \in \mathbf{I}_{\text{pop}}} \mathbf{v}_i^s. \tag{3}$$

The model update of each fake user of the targeted item $t$ at round $l$ can be now computed as $g_t^l = \mathbf{v}_t' - \mathbf{v}_t^l$, where $\mathbf{v}_t^l$ is the item embedding of the targeted item $t$ in the $l$-th global model.

### 4.2.3 Select filler items.
In real-world recommender systems, genuine users typically evaluate a subset of items. In our proposed attack, to mimic the rating behavior of these genuine users and avoid future detection, each fake user rates not only the targeted item but also certain chosen items, which we call *filler items*. In our experiments, we also find that the hit ratio of the targeted item will drop gradually if each fake user interacts with only the targeted item. This is because the target model $\mathbf{v}_t'$ is fixed and fake users do not operate items other than the targeted item. As a result, genuine users will increase the ratings of their positive samples to make them rank higher than the targeted item. Consequently, the target hit ratio will decrease. To mitigate this decline, each fake user can employ filler items, ensuring the target hit ratio decreases at an even slower rate and therefore maintaining a high target hit ratio. For those filler items, we hope their predicted scores won't change too much so the targeted item can maintain a high ranking. Our approach is: to record the initial item features of filler items right before the first attacking round, and in each following global round, the target features of filler items are set to their recorded features.

We can further define this process. Let's say each fake user has the option to select $f$ items as filler items, distinct from the targeted item. These are chosen based on their deviation from the original embeddings when the fake user initiates an attack. Consider that fake users begin their attack in the $s$-th global round. At the start of that round, they record the item embeddings, represented as:

$$\mathbf{V}^s = \{\mathbf{v}_1^s, \mathbf{v}_2^s, \ldots, \mathbf{v}_m^s\}.$$

In the $l$-th global round, each fake user calculates the deviation as:

$$d_i = \|\mathbf{v}_i^s - \mathbf{v}_i^l\|_2, i = 1, 2, \cdots, m$$

The fake users then rank the $d_i$ values in descending order and choose the $f$ filler items with the largest $d_i$. These selected items must exclude the targeted item $t$. If the targeted item is included, it is removed. Denote the set of filler items as $\mathcal{F}$. The model update for filler item $i \in \mathcal{F}$ is then given by:

$$\mathbf{g}_i^l = \mathbf{v}_i^l - \mathbf{v}_i^s.$$

Note that the filler items of each fake user may vary in different global rounds. Therefore, the total number of filler items in all global rounds may be larger than the number of filler items chosen in each single round. These fake users may be detected if they interact with too many items in total. We conduct an experiment where we record filler items chosen in each global round by the first fake user to implement the attack. We set the proportion of fake users to be 0.05%, where the Yelp dataset and Median [43] aggregation rule are considered. After calculating the union of filler items chosen in each attack round, we find that fake users only choose 83 items


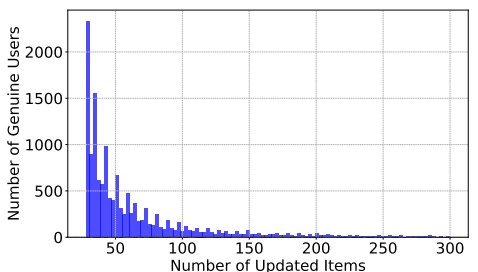

**Figure 2: Distribution of the number of updated items for genuine users.**

in total. A primary reason for this limited variety is the repeat selection of certain items, especially popular ones, as filler items in numerous attack rounds. For genuine users, the number of item model updates (including items of negative samples) ranges from 28 to 2046, with the mean $\mu = 76$ and the standard deviation $\sigma = 95$. Figure 2 shows the distribution of the number of updated items for genuine users. According to Figure 2, we can ensure that, within our experimental parameters, fake users remain undetectable based on the volume of their overall filler item interactions.

*4.2.4 Sending malicious model updates.* From the above procedures, each fake user has computed the model update of the targeted item in the $l$-th global round $\mathbf{g}_t^l$ as well as model updates of filler items $\mathbf{g}_i^l$ in which $i \in \mathcal{F}$, where $\mathcal{F}$ is the set of filler items. At the end of the global round, each fake user sends $\mathbf{g}_t^l$ and each $\mathbf{g}_i^l (i \in \mathcal{F})$ computed above to the server.

**Complete algorithm:** Algorithm 3 summarizes the complete algorithm of our proposed PoisonFRS attack. Note that in our PoisonFRS attack, the fake users start to attack the recommender systems from the global training round $s$, i.e., these fake users do not join the training process until the $s$-th training round. Lines 3-8 of Algorithm 3 is the process of estimating $k$ popular items. The $\mathbf{I}_{\text{pop}} = \{i_1, i_2, \ldots, i_k\}$ in Line 8 is computed to record the indices of the selected items. In lines 9-10, the attacker computes the targeted item embedding of the target model. In lines 11-16, the attacker records all items' embedding of the first attack round and selects filler items. In lines 17-19, the attacker computes the model update and uploads it to the server.

## 5 EXPERIMENTS

### 5.1 Experimental Setup

*5.1.1 Datasets.* In our experiments, we use four real-world datasets to evaluate the effectiveness of our proposed PoisonFRS attack. These datasets are Steam-200K (Steam) [7], Yelp [42], MovieLens-10M (ML-10M) [13] and MovieLens-20M (ML-20M) [13]. These datasets come from multiple domains and their sizes vary from small to large. For example, Steam is a dataset about user interactions on Steam, which has 3,753 users and 5,134 items with 114,713 interactions, while ML-20M is a large dataset from GroupLens with about 20,000,263 ratings of 138,493 users on 26,740 movies. In each dataset, we split the last item that each user interacts with into the test set. Table 2 shows the detailed statistics of four datasets.

*5.1.2 Compared attacks.* We compare our proposed PoisonFRS with five traditional poisoning attacks (Random [12], Popular [12],

---

**Algorithm 3** Our PoisonFRS Attack.

**Input:** Number of global rounds $T$, number of items $m$, attack starting round $s$, number of filler items $f$, number of popular items $k$, scaling factor $\lambda$.

**Output:** Targeted item model update $g_t^l$ and filler item model updates $\mathbf{g}_i^l$

1: **for** $l = 1, 2, \ldots, T$ **do**
2:   **if** $l = s$ **then**
3:     $\mathbf{v}_{\text{avg}} \leftarrow \frac{1}{m} \sum_{i=1}^{m} \mathbf{v}_i^s$
4:     **for** $i = 1, 2, \ldots, m$ **do**
5:       Compute inner product $p_i \leftarrow \langle \mathbf{v}_i^s, \mathbf{v}_{\text{avg}} \rangle$
6:     **end for**
7:     Sort as $p_{i_1} \leq p_{i_2} \leq \cdots \leq p_{i_m}$
8:     $\mathbf{I}_{\text{pop}} \leftarrow \{i_1, i_2, \cdots, i_k\}$
9:     $\mathbf{v}_t \leftarrow \frac{1}{k} \sum_{i \in \mathbf{I}_{\text{pop}}} \mathbf{v}_i^s$
10:    $\mathbf{v}_t' \leftarrow \frac{\lambda}{k} \sum_{i \in \mathbf{I}_{\text{pop}}} \mathbf{v}_i^s$
11:    $\mathbf{V}^s \leftarrow \{\mathbf{v}_1^s, \mathbf{v}_2^s, \ldots, \mathbf{v}_m^s\}$
12:   **end if**
13:   **if** $l \geq s$ **then**
14:    $g_t^l \leftarrow \mathbf{v}_t' - \mathbf{v}_t^l$
15:    $d_i \leftarrow \|\mathbf{v}_i^s - \mathbf{v}_i^l\|_2$ for all $i$
16:    Select $f$ filler items with largest $d_i$, denoted the index set of filler items as $\mathcal{F}$
17:    $\mathbf{g}_t^l \leftarrow \mathbf{v}_t^l - \mathbf{v}_t$
18:    $\mathbf{g}_i^l \leftarrow \mathbf{v}_i^l - \mathbf{v}_i^s$ for each $i \in \mathcal{F}$
19:    Upload $\mathbf{g}_t^l$ and $\mathbf{g}_i^l$ to the server
20:   **end if**
21: **end for**

---

**Table 2: Statistics of datasets.**

| Dataset | # Users | # Items | # Ratings |
|---------|---------|---------|-----------|
| Steam | 3,753 | 5,134 | 114,713 |
| Yelp | 14,575 | 25,602 | 569,947 |
| ML-10M | 69,878 | 10,673 | 10,000,054 |
| ML-20M | 138,493 | 26,740 | 20,000,263 |

Bandwagon [16], RAPU-G [47], RAPU-R [47]) and three state-of-the-art poisoning attacks on FedRecs (FedRecAttack [29], PipAttack [48], PSMU [45]).

**Random [12]:** This is a simple attack performed on recommender systems. The attacker chooses the targeted item and other random items as filler items. Initially designed for centralized recommender systems, it can be transferred to FedRecs: the attacker constructs fake users according to the above method, and those fake users do regular training.

**Popular [12]:** Like Random attack, Popular attack is initially designed for centralized recommender systems. The difference between Popular and Random attacks is that in Popular attack, the attacker chooses the most popular items as filler items.

**Bandwagon [16]:** The difference between Bandwagon attack and Popular attack is that the attacker does not set all filler as popular items in Bandwagon attack. Instead, it only puts a proportion of filler items (in our experiments, 10%) to be popular items, while other filler items are randomly chosen from the remaining unselected items.

**RAPU-G [47]:** In RAPU-G attack, the attacker uses a probabilistic generative model [44] to identify unperturbed user and item interaction data, which is then utilized to create fake user-item interactions.

**RAPU-R [47]:** In RAPU-R attack, the attacker takes a different approach by reversing the learning process of the model and incorporating heuristic rules specific to the context. This attack is also initially designed for centralized recommender systems.

**FedRecAttack [29]:** FedRecAttack is a targeted poisoning attack designed for FedRecs. It requires the attacker to have partial interaction data. With those data, the attacker can estimate genuine user features. In this way, the attacker can optimize its loss function about the popularity of the targeted item.

**PipAttack [48]:** This is an attack tailored to FedRecs. In PipAttack, the attacker requires knowledge about the popularity of each item. With such knowledge, it can construct a popularity estimator to predict the popularity given an item feature. Therefore, it can generate model updates that drive the item toward high popularity.

**PSMU [45]:** In PSMU attack, the attacker randomly generates local training data for each fake user at each attacking round. After that, each user trains the local model using its local training data and gets a user feature. PSMU assumes that if the targeted item is popular with fake users, it is also popular with other users. Therefore, the attacker optimizes a loss function to enlarge the popularity of the targeted item among fake users.

We note that there are some prevailing attacks on centralized recommender systems [9, 20, 20, 32, 34, 37, 38, 40, 46, 49], but they are concentrated on explicit feedback and cannot be adapted to our implicit feedback setting. Therefore, we do not adopt these attacks as baselines.

*5.1.3 Aggregation rules.* In our experiments, we consider the following aggregation rules.

**FedAvg [24]:** In FedAvg, upon the server receives local model updates from all users, it computes the average of these received model updates.

**Coordinate-wise median (Median) [43]:** Median serves as an aggregation rule that operates on an individual dimension basis. After gathering model updates from all users, the server computes the median value for each dimension. This approach inherently mitigates the potential impact of outlier updates, ensuring the resilience of the global model against extreme values that might represent malicious alterations.

**Coordinate-wise trimmed mean (Trimmed-mean) [43]:** Trimmed-mean is also a coordinate-wise aggregation rule. For each dimension, the server first removes the largest $\beta$ and the smallest $\beta$ values in all collected model updates, then computes the average of the remaining elements as the corresponding parameter in the global model update, where $\beta$ is the trimmed parameter.

**Krum [4]:** Suppose there are $n$ users, with $m$ being malicious/fake. Under the Krum aggregation rule, each user $i$ selects $n - m - 2$ users whose model updates are closest. Then Krum calculates the user's score as the average $\ell_2$ distance from $g_i^t$ to its closest neighbors' vectors $g_j^t$. Clients closer to their nearest $n - m - 2$ neighbors

receive lower scores. Assuming the benign user is very close to its neighbors, we assign $g^t$ as $g_{i*}^t$, where $g_{i*}^t$ has the smallest score.

**Clip [17]:** In this method, the $\ell_2$ norm of the model update of each user is limited within a bound. Model updates whose $\ell_2$ norm surpasses the bound will be scaled to be within the bound. The clipped parameter is set to 3 in our experiments.

**FLAME [26]:** In FLAME, the server first computes the cosine similarity between each two model updates and generates a distance matrix. After that, it leverages HDBSCAN [5] clustering method to cluster those model updates, setting min_cluster_size $= n/2 + 1$ and min_samples $= 1$, thus chooses a cluster whose members are most likely to be genuine, where $n$ is the total number of users. Finally, it adaptively clips collected model updates and computes the average of clipped model updates.

**HiCS [45]:** This approach forms a gradient bank to accumulate collected model updates. In each global round, it accumulates received model updates in the gradient bank and then chooses the top-$z$ largest elements in the bank and subtracts them from the bank (gradient sparsification). Then, the server adaptively clips them based on their average magnitude. After that, the server computes the average of these clipped model updates.

*5.1.4 Parameter setting.* The parameter $\lambda$ is set to 10 in all datasets. $k$ is set to 5. The attacker starts to attack at the 50-th global round. We set the number of global rounds to 300 to ensure the model converges. The number of filler items in our proposed PoisonFRS and all baseline attacks is 59. The learning rates for all datasets are 0.05. In our paper, the most unpopular item (the item with the least number of rating scores) is chosen as the targeted item.

## 5.2 Experimental Results

**Our attack significantly outperforms all baseline attacks:** We tested the attack effect of our method and baselines on seven aggregation rules. The results of FedAvg and Median aggregation rules are shown in Table 3, and the results of the other five aggregation rules are shown in Table 6 in Appendix. "Attack size" denotes the fraction of fake users. "None" represents the setting without attack. We can observe from Table 3 and Table 6 that centralized recommender system-based attacks almost show no attacking effect, which means attacks tailored to FedRecs are quite needed. For attacks on FedRecs, FedRecAttack shows the best result among baselines. This is because such an attack requires the most prior knowledge – partial of the raw interaction matrix. Then is the PipAttack because the attacker in PipAttack knows the exact popularity of each item. PSMU shows the worst effect in our experiment because the datasets are quite sparse, and the attacker cannot get the local training data consistent with benign users by randomly choosing rated items. All these baselines, including FedRecAttack, fail when the proportion of fake users is extremely small, like 0.03%. However, in our proposed PoisonFRS attack, the hit ratio of the targeted item significantly surpasses all baselines, and when the proportion of fake users is really small, our attack still shows a strong effect.

**Our attack can break current defenses:** From Table 3 and Table 6, we conclude that for most baseline attacks, their effect will be weakened to some extent when facing defensive aggregation rules.

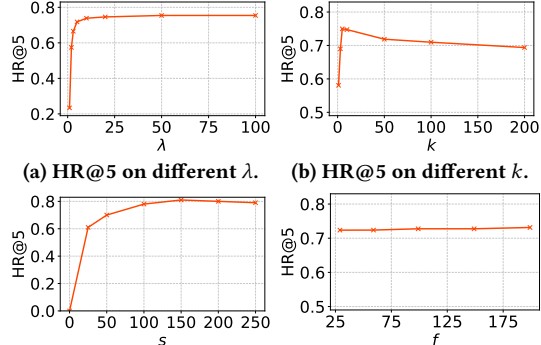

(a) HR@5 on different $\lambda$.     (b) HR@5 on different $k$.

(c) HR@5 on different $s$.     (d) HR@5 on different $f$.

**Figure 3: Result of ablation studies on Yelp dataset, where FedAvg aggregation rule is considered.**

However, for our attack, we can see its effect is almost aggregation rule agnostic. Our attack is strongly effective even under HiCS, the defense tailored to FedRecs. Therefore, the attacker can achieve its goal regardless of the server's aggregation rule.

**Table 3: HR@5 for different attacks under FedAvg and Median aggregation rules.**

(a) FedAvg

| Dataset | Attack size | None | Rand-om | Popu-lar | Band-wagon | RAPU-G | RAPU-R | FedRec-Attack | Pip-Attack | PSMU | Poison-FRS |
|---|---|---|---|---|---|---|---|---|---|---|---|
| Steam | 0.03% | 0.00 | 0.00 | 0.00 | 0.00 | 0.00 | 0.00 | 0.00 | 0.00 | 0.00 | **0.98** |
| | 0.05% | 0.00 | 0.00 | 0.00 | 0.00 | 0.00 | 0.00 | 0.00 | 0.00 | 0.00 | **0.99** |
| | 0.1% | 0.00 | 0.00 | 0.00 | 0.00 | 0.00 | 0.00 | 0.91 | 0.00 | 0.00 | **0.99** |
| | 0.5% | 0.00 | 0.00 | 0.03 | 0.00 | 0.00 | 0.00 | 0.89 | 0.01 | 0.00 | **0.99** |
| Yelp | 0.03% | 0.00 | 0.00 | 0.00 | 0.00 | 0.00 | 0.00 | 0.19 | 0.00 | 0.00 | **0.72** |
| | 0.05% | 0.00 | 0.00 | 0.00 | 0.00 | 0.00 | 0.00 | 0.20 | 0.00 | 0.00 | **0.73** |
| | 0.1% | 0.00 | 0.00 | 0.00 | 0.00 | 0.00 | 0.00 | 0.22 | 0.00 | 0.00 | **0.76** |
| | 0.5% | 0.00 | 0.00 | 0.02 | 0.00 | 0.00 | 0.00 | 0.32 | 0.12 | 0.00 | **0.77** |
| ML-10M | 0.03% | 0.00 | 0.00 | 0.00 | 0.00 | 0.00 | 0.00 | 0.00 | 0.00 | 0.00 | **0.99** |
| | 0.05% | 0.00 | 0.00 | 0.00 | 0.00 | 0.00 | 0.00 | 0.00 | 0.00 | 0.00 | **1.00** |
| | 0.1% | 0.00 | 0.00 | 0.00 | 0.00 | 0.00 | 0.00 | 0.00 | 0.00 | 0.00 | **1.00** |
| | 0.5% | 0.00 | 0.00 | 0.00 | 0.00 | 0.00 | 0.00 | 0.01 | 0.00 | 0.00 | **1.00** |
| ML-20M | 0.03% | 0.00 | 0.00 | 0.00 | 0.00 | 0.00 | 0.00 | 0.00 | 0.00 | 0.00 | **0.99** |
| | 0.05% | 0.00 | 0.00 | 0.00 | 0.00 | 0.00 | 0.00 | 0.00 | 0.00 | 0.00 | **0.99** |
| | 0.1% | 0.00 | 0.00 | 0.00 | 0.00 | 0.00 | 0.00 | 0.00 | 0.00 | 0.00 | **0.99** |
| | 0.5% | 0.00 | 0.00 | 0.00 | 0.00 | 0.00 | 0.00 | 0.00 | 0.00 | 0.00 | **0.99** |

(b) Median

| Dataset | Attack size | None | Rand-om | Popu-lar | Band-wagon | RAPU-G | RAPU-R | FedRec-Attack | Pip-Attack | PSMU | Poison-FRS |
|---|---|---|---|---|---|---|---|---|---|---|---|
| Steam | 0.03% | 0.00 | 0.00 | 0.00 | 0.00 | 0.00 | 0.00 | 0.00 | 0.00 | 0.00 | **0.98** |
| | 0.05% | 0.00 | 0.00 | 0.00 | 0.00 | 0.00 | 0.00 | 0.00 | 0.00 | 0.00 | **0.98** |
| | 0.1% | 0.00 | 0.00 | 0.00 | 0.00 | 0.00 | 0.00 | 0.89 | 0.00 | 0.00 | **0.99** |
| | 0.5% | 0.00 | 0.00 | 0.03 | 0.00 | 0.00 | 0.00 | 0.89 | 0.02 | 0.00 | **0.99** |
| Yelp | 0.03% | 0.00 | 0.00 | 0.00 | 0.00 | 0.00 | 0.00 | 0.23 | 0.00 | 0.00 | **0.71** |
| | 0.05% | 0.00 | 0.00 | 0.00 | 0.00 | 0.00 | 0.00 | 0.26 | 0.00 | 0.00 | **0.70** |
| | 0.1% | 0.00 | 0.00 | 0.00 | 0.00 | 0.00 | 0.00 | 0.63 | 0.00 | 0.00 | **0.72** |
| | 0.5% | 0.00 | 0.00 | 0.05 | 0.00 | 0.01 | 0.01 | 0.31 | 0.14 | 0.00 | **0.79** |
| ML-10M | 0.03% | 0.00 | 0.00 | 0.00 | 0.00 | 0.00 | 0.00 | 0.00 | 0.00 | 0.00 | **0.38** |
| | 0.05% | 0.00 | 0.00 | 0.00 | 0.00 | 0.00 | 0.00 | 0.00 | 0.00 | 0.00 | **0.99** |
| | 0.1% | 0.00 | 0.00 | 0.00 | 0.00 | 0.00 | 0.00 | 0.00 | 0.00 | 0.00 | **0.99** |
| | 0.5% | 0.00 | 0.00 | 0.00 | 0.00 | 0.00 | 0.00 | 0.00 | 0.00 | 0.00 | **0.99** |
| ML-20M | 0.03% | 0.00 | 0.00 | 0.00 | 0.00 | 0.00 | 0.00 | 0.00 | 0.00 | 0.00 | **0.99** |
| | 0.05% | 0.00 | 0.00 | 0.00 | 0.00 | 0.00 | 0.00 | 0.00 | 0.00 | 0.00 | **0.99** |
| | 0.1% | 0.00 | 0.00 | 0.00 | 0.00 | 0.00 | 0.00 | 0.00 | 0.00 | 0.00 | **0.99** |
| | 0.5% | 0.00 | 0.00 | 0.00 | 0.00 | 0.00 | 0.00 | 0.00 | 0.00 | 0.00 | **0.99** |

**Impact of different $\lambda$:** In our method, the parameter $\lambda$ represents the amplification factor applied to the targeted item features to enhance the influence of the target rating scores among genuine users. Intuitively, a larger $\lambda$ is often deemed advantageous, and as $\lambda$ surpasses a certain threshold, the effectiveness of the attack saturates. In this part, we investigate the impact of varying $\lambda$ values, specifically setting $\lambda$ to the following values: 1, 2, 3, 5, 10, 20, 50, and 100. We then assess the resulting target hit ratios of Yelp under FedAvg when the proportion of fake users is fixed at 0.05%. Figure 3(a) displays our experimental results. It is evident that when $\lambda$ is set to 1, the target hit ratio is notably low. This is attributed to the fact that the predicted score of the targeted item does not stand out sufficiently. As $\lambda$ increases, the target hit ratio rises, ultimately reaching saturation at approximately $\lambda = 10$. In practice, where the attacker may not have prior knowledge of the ideal $\lambda$ value, choosing a sufficiently large $\lambda$ is recommended, as the attack effect tends to saturate under such conditions.

**Impact of different $k$:** Our attack has a parameter $k$, which represents the number of popular items the attacker chooses to construct the target model. Usually, $k$ cannot either be too large or too small: if $k$ is too large, then some unpopular items will be included, and the constructed target model will not cause a very high hit ratio on the targeted item; if $k$ is too small, then the chosen item features are insufficient to cover all features that gain popularity in a majority of users. To explore the precise impact on different $k$, we set $k$ to be 1, 3, 5, 10, 50, 100, and 200 and measure the hit ratio of the targeted item, respectively. Note that when $k$ is larger than 5, the optimal value of $\lambda$ will increase because the magnitude of the average of $k$ item features will be smaller (some elements may counteract), and in this way, we cannot say the hit ratio decreases because of the increase of $k$. To address this problem, we set $\lambda = 100$ to ensure that the attack effect saturates in the aspect of $\lambda$ and is only influenced by the choice of $k$. In this experiment, we still test Yelp under FedAvg with 0.05% fake users. Figure 3(b) shows our result. From the figure, we can see that when $k$ is set to 1, the hit ratio of the targeted item is below 0.6. However, when $k$ increases to 5, the hit ratio reaches the peak—about 0.75. After $k$ continues to increase, the target hit ratio decreases instead. Although the attack effect is relative to the choice of $k$, the attacker need not worry about this – from Figure 3(b), we can see that the attack result is very high when $k$ is in a wide range of 1 to 200. This means it is enough for the attacker to just set $k$ to a relatively reasonable value, and the attack effect will be satisfying.

**Impact of different $s$:** In our PoisonFRS, the attacker has the flexibility to initiate the attack at any global round. In our default experimental setup, the attacker commences the attack from the 50-th global training round. This section explores the impact of varying attack initiation times, specifically considering scenarios where the attack starts after 0, 25, 50, 100, 150, 200, and 250 global rounds. This experiment aims to assess how the timing of the attack initiation influences the effectiveness of our proposed attack. Our dataset for this experiment is Yelp. It is tested under FedAvg with 0.05% fake users.

The results are presented in Figure 3(c). From the results, it becomes evident that the performance of our attack improves when the attacker initiates the attack later, such as during the 150-th or

200-th round. This improvement can be attributed to the fact that delaying the attack allows the attacker to gather more information about the items. Consequently, the attacker can make more precise estimations of popular items and construct a more effective targeted item embedding using Eq. (2) and Eq. (3). However, if the attacker starts the attack too late, there may not be sufficient time to have a significant impact. Hence, choosing a suitable attack initiation time is crucial for success.

**Impact of different $f$:** In our attack, fake users also select some filler items. To delve deeper into the influence of the number of filler items, we conducted experiments by setting $f$ to various values: 0, 29, 59, 99, 149, and 199. This range corresponds to varying the overall number of interacted items per fake user, spanning from 1 to 200. We employed the Yelp dataset with a proportion of fake users set at 0.05%. The results are presented in Figure 3(d). From Figure 3(d), we can observe that the choice of $f$ hardly influences the attacking effect. However, a suitable $f$ can make our attack more stealthy and prevent fake users from being detected.

**Adding noise to the malicious model updates:** In our method, the target model is fixed. Therefore, if two or more fake users attack at the same global round, they will send the same model update. It is likely to happen when the proportion of fake users is high, and as a result, the server may detect it by finding that the model updates of these fake users are the same. We can address this issue by adding random Gaussian noise to each malicious model update. However, it is not clear whether this influences the attacking effect. To further explore its practicability, we add Gaussian noise $\mathcal{N}(\mathbf{0}, \mathbf{I})$ to each item's malicious model update and test the attacking effect. We choose Yelp and FedAvg as the aggregation rules to conduct our experiment. Table 4 shows the target hit ratio under this setting when the proportion of fake users ranges from 0.03% to 0.5%. Comparing Table 4 with Table 3(a), we can see that the effect is almost no different from the default setting. This experiment further demonstrates the robustness of our algorithm, that it is resilient to a reasonable magnitude of Gaussian noise perturbations.

**Table 4: Hit ratio of the targeted item after adding noise to the malicious model update, where Yelp dataset and FedAvg aggregation rule are considered.**

| Attack size | 0.03% | 0.05% | 0.1% | 0.5% |
|---|---|---|---|---|
| HR@5 | 0.72 | 0.73 | 0.76 | 0.79 |

**Results on different metrics:** In the default setting, we employ HR@5 as our primary evaluation metric for assessing the attack's impact. However, we also explore additional metrics such as HR@10, HR@50, and normalized discounted cumulative gain (NDCG) to ensure a comprehensive assessment of our method's performance against various attacks. The results are presented in Table 7 in Appendix. The table shows that our method consistently outperforms the baselines across all these metrics. These findings establish the general superiority of our approach over the baselines across a diverse set of evaluation metrics.

**Results on larger attack size:** In the default setting, the proportion of fake users is minimal, and most baseline methods exhibit

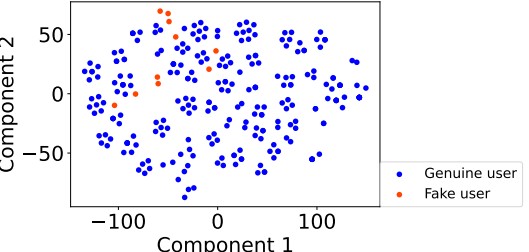

**Figure 4: Genuine and fake users in the latent space.**

weak effects under these conditions. To assess the continued superiority of our method over baseline approaches in situations with a larger proportion of fake users, we vary the attack size to 1%, 3%, and 10%. The results are summarized in Table 5. From Table 5, it is evident that as the attack size increases, the impact of the baseline methods also becomes more pronounced. However, even in these scenarios, our method consistently outperforms the baseline methods, demonstrating superior performance.

**Detection results:** To ensure that our proposed PoisonFRS attack remains undetected by the server, we experimented to determine whether the server can discern the targeted item embedding update from genuine users and fake users. We employed t-SNE [35] for dimensionality reduction and visualization. The results depicted in Figure 4 illustrate that the targeted item embedding updates from genuine users and fake users are intermingled to such an extent that our attack becomes exceedingly difficult to detect.

**Table 5: HR@5 of larger attack sizes. Yelp dataset and FedAvg aggregation rule are considered.**

| Attack size | None | Random | Popular | Bandwagon | RAPU-G | RAPU-R | FedRec-Attack | Pip-Attack | PSMU | Poison-FRS |
|---|---|---|---|---|---|---|---|---|---|---|
| 1% | 0.00 | 0.00 | 0.06 | 0.00 | 0.03 | 0.00 | 0.59 | 0.18 | 0.00 | **0.75** |
| 5% | 0.00 | 0.01 | 0.18 | 0.43 | 0.34 | 0.07 | 0.48 | 0.16 | 0.00 | **0.80** |
| 10% | 0.00 | 0.01 | 0.23 | 0.62 | 0.44 | 0.18 | 0.56 | 0.24 | 0.00 | **0.81** |

## 6 CONCLUSION

In this paper, we have identified certain limitations in current attacks targeting FedRecs. These limitations stem from the requirement for information from genuine users or access to local training data, which can pose significant challenges, especially for recently registered fake users. Furthermore, these attacks have been proven to be ineffective when the proportion of fake users is extremely low. Motivated by these observations, we have introduced a novel poisoning attack aimed at FedRecs, using fake users. In our proposed attack, fake users neither possess local training data nor have information about the genuine users. Through comprehensive experiments conducted on four distinct datasets, we have demonstrated that by injecting a small percentage of fake users, our attack can successfully promote the targeted item to a vast majority of genuine users, and when defenses specifically designed for FedRecs are deployed. As a result of our findings, an interesting future research lies in exploring defense mechanisms that can effectively withstand the attack we have introduced.

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

## Table 6: HR@5 for different attacks under Trimmed-mean, Clip, Krum, FLAME, and HiCS aggregation rules.

**(a) Trimmed-mean**

| Dataset | Attack size | None | Random | Popular | Bandwagon | RAPU-G | RAPU-R | FedRec-Attack | Pip-Attack | PSMU | Poison-FRS |
|---|---|---|---|---|---|---|---|---|---|---|---|
| Steam | 0.03% | 0.00 | 0.00 | 0.00 | 0.00 | 0.00 | 0.00 | 0.00 | 0.00 | 0.00 | **0.98** |
| | 0.05% | 0.00 | 0.00 | 0.00 | 0.00 | 0.00 | 0.00 | 0.00 | 0.00 | 0.00 | **0.99** |
| | 0.1% | 0.00 | 0.00 | 0.00 | 0.00 | 0.00 | 0.00 | 0.87 | 0.00 | 0.00 | **0.99** |
| | 0.5% | 0.00 | 0.00 | 0.06 | 0.00 | 0.00 | 0.00 | 0.88 | 0.01 | 0.00 | **0.99** |
| Yelp | 0.03% | 0.00 | 0.00 | 0.00 | 0.00 | 0.00 | 0.00 | 0.19 | 0.00 | 0.00 | **0.72** |
| | 0.05% | 0.00 | 0.00 | 0.00 | 0.00 | 0.00 | 0.00 | 0.20 | 0.00 | 0.00 | **0.73** |
| | 0.1% | 0.00 | 0.00 | 0.00 | 0.00 | 0.00 | 0.00 | 0.22 | 0.00 | 0.00 | **0.74** |
| | 0.5% | 0.00 | 0.00 | 0.00 | 0.00 | 0.01 | 0.00 | 0.34 | 0.12 | 0.00 | **0.74** |
| ML-10M | 0.03% | 0.00 | 0.00 | 0.00 | 0.00 | 0.00 | 0.00 | 0.00 | 0.00 | 0.00 | **0.98** |
| | 0.05% | 0.00 | 0.00 | 0.00 | 0.00 | 0.00 | 0.00 | 0.00 | 0.00 | 0.00 | **0.99** |
| | 0.1% | 0.00 | 0.00 | 0.00 | 0.00 | 0.00 | 0.00 | 0.00 | 0.00 | 0.00 | **0.99** |
| | 0.5% | 0.00 | 0.00 | 0.00 | 0.00 | 0.00 | 0.00 | 0.00 | 0.00 | 0.00 | **1.00** |
| ML-20M | 0.03% | 0.00 | 0.00 | 0.00 | 0.00 | 0.00 | 0.00 | 0.00 | 0.00 | 0.00 | **0.98** |
| | 0.05% | 0.00 | 0.00 | 0.00 | 0.00 | 0.00 | 0.00 | 0.00 | 0.00 | 0.00 | **0.99** |
| | 0.1% | 0.00 | 0.00 | 0.00 | 0.00 | 0.00 | 0.00 | 0.00 | 0.00 | 0.00 | **0.99** |
| | 0.5% | 0.00 | 0.00 | 0.00 | 0.00 | 0.00 | 0.00 | 0.02 | 0.00 | 0.00 | **0.99** |

**(b) Clip**

| Dataset | Attack size | None | Random | Popular | Bandwagon | RAPU-G | RAPU-R | FedRec-Attack | Pip-Attack | PSMU | Poison-FRS |
|---|---|---|---|---|---|---|---|---|---|---|---|
| Steam | 0.03% | 0.00 | 0.00 | 0.00 | 0.00 | 0.00 | 0.00 | 0.00 | 0.00 | 0.00 | **0.98** |
| | 0.05% | 0.00 | 0.00 | 0.00 | 0.00 | 0.00 | 0.00 | 0.01 | 0.00 | 0.00 | **0.99** |
| | 0.1% | 0.00 | 0.00 | 0.00 | 0.00 | 0.00 | 0.00 | 0.91 | 0.00 | 0.00 | **0.99** |
| | 0.5% | 0.00 | 0.00 | 0.01 | 0.00 | 0.00 | 0.00 | 0.89 | 0.00 | 0.00 | **0.99** |
| Yelp | 0.03% | 0.00 | 0.00 | 0.00 | 0.00 | 0.00 | 0.00 | 0.19 | 0.00 | 0.00 | **0.69** |
| | 0.05% | 0.00 | 0.00 | 0.00 | 0.00 | 0.00 | 0.00 | 0.18 | 0.00 | 0.00 | **0.72** |
| | 0.1% | 0.00 | 0.00 | 0.00 | 0.00 | 0.00 | 0.00 | 0.19 | 0.02 | 0.00 | **0.73** |
| | 0.5% | 0.00 | 0.00 | 0.02 | 0.00 | 0.00 | 0.00 | 0.31 | 0.03 | 0.00 | **0.74** |
| ML-10M | 0.03% | 0.00 | 0.00 | 0.00 | 0.00 | 0.00 | 0.00 | 0.00 | 0.00 | 0.00 | **0.19** |
| | 0.05% | 0.00 | 0.00 | 0.00 | 0.00 | 0.00 | 0.00 | 0.00 | 0.00 | 0.00 | **0.99** |
| | 0.1% | 0.00 | 0.00 | 0.00 | 0.00 | 0.00 | 0.00 | 0.00 | 0.00 | 0.00 | **0.99** |
| | 0.5% | 0.00 | 0.00 | 0.00 | 0.00 | 0.00 | 0.00 | 0.01 | 0.00 | 0.00 | **0.99** |
| ML-20M | 0.03% | 0.00 | 0.00 | 0.00 | 0.00 | 0.00 | 0.00 | 0.00 | 0.00 | 0.00 | **0.99** |
| | 0.05% | 0.00 | 0.00 | 0.00 | 0.00 | 0.00 | 0.00 | 0.00 | 0.00 | 0.00 | **0.99** |
| | 0.1% | 0.00 | 0.00 | 0.00 | 0.00 | 0.00 | 0.00 | 0.00 | 0.00 | 0.00 | **0.99** |
| | 0.5% | 0.00 | 0.00 | 0.00 | 0.00 | 0.00 | 0.00 | 0.00 | 0.00 | 0.00 | **0.99** |

**(c) Krum**

| Dataset | Attack size | None | Random | Popular | Bandwagon | RAPU-G | RAPU-R | FedRec-Attack | Pip-Attack | PSMU | Poison-FRS |
|---|---|---|---|---|---|---|---|---|---|---|---|
| Steam | 0.03% | 0.00 | 0.00 | 0.00 | 0.00 | 0.00 | 0.00 | 0.00 | 0.00 | 0.00 | **0.98** |
| | 0.05% | 0.00 | 0.00 | 0.00 | 0.00 | 0.00 | 0.00 | 0.00 | 0.00 | 0.00 | **0.98** |
| | 0.1% | 0.00 | 0.00 | 0.00 | 0.00 | 0.00 | 0.00 | 0.89 | 0.00 | 0.00 | **0.99** |
| | 0.5% | 0.00 | 0.00 | 0.06 | 0.00 | 0.00 | 0.00 | 0.89 | 0.01 | 0.00 | **0.99** |
| Yelp | 0.03% | 0.00 | 0.00 | 0.00 | 0.00 | 0.00 | 0.00 | 0.33 | 0.00 | 0.00 | **0.71** |
| | 0.05% | 0.00 | 0.00 | 0.00 | 0.00 | 0.00 | 0.00 | 0.19 | 0.00 | 0.00 | **0.68** |
| | 0.1% | 0.00 | 0.00 | 0.00 | 0.00 | 0.00 | 0.00 | 0.33 | 0.00 | 0.00 | **0.75** |
| | 0.5% | 0.00 | 0.00 | 0.06 | 0.00 | 0.01 | 0.00 | 0.38 | 0.11 | 0.00 | **0.76** |
| ML-10M | 0.03% | 0.00 | 0.00 | 0.00 | 0.00 | 0.00 | 0.00 | 0.00 | 0.00 | 0.00 | **0.98** |
| | 0.05% | 0.00 | 0.00 | 0.00 | 0.00 | 0.00 | 0.00 | 0.00 | 0.00 | 0.00 | **1.00** |
| | 0.1% | 0.00 | 0.00 | 0.00 | 0.00 | 0.00 | 0.00 | 0.00 | 0.00 | 0.00 | **1.00** |
| | 0.5% | 0.00 | 0.00 | 0.00 | 0.00 | 0.00 | 0.00 | 0.00 | 0.00 | 0.00 | **1.00** |
| ML-20M | 0.03% | 0.00 | 0.00 | 0.00 | 0.00 | 0.00 | 0.00 | 0.00 | 0.00 | 0.00 | **0.99** |
| | 0.05% | 0.00 | 0.00 | 0.00 | 0.00 | 0.00 | 0.00 | 0.00 | 0.00 | 0.00 | **0.99** |
| | 0.1% | 0.00 | 0.00 | 0.00 | 0.00 | 0.00 | 0.00 | 0.00 | 0.00 | 0.00 | **0.99** |
| | 0.5% | 0.00 | 0.00 | 0.00 | 0.00 | 0.00 | 0.00 | 0.00 | 0.00 | 0.00 | **0.99** |

**(d) FLAME**

| Dataset | Attack size | None | Random | Popular | Bandwagon | RAPU-G | RAPU-R | FedRec-Attack | Pip-Attack | PSMU | Poison-FRS |
|---|---|---|---|---|---|---|---|---|---|---|---|
| Steam | 0.03% | 0.00 | 0.00 | 0.00 | 0.00 | 0.00 | 0.00 | 0.00 | 0.00 | 0.00 | **0.98** |
| | 0.05% | 0.00 | 0.00 | 0.00 | 0.00 | 0.00 | 0.00 | 0.00 | 0.00 | 0.00 | **0.98** |
| | 0.1% | 0.00 | 0.00 | 0.00 | 0.00 | 0.00 | 0.00 | 0.00 | 0.00 | 0.00 | **0.99** |
| | 0.5% | 0.00 | 0.00 | 0.00 | 0.00 | 0.00 | 0.00 | 0.90 | 0.00 | 0.00 | **0.99** |
| Yelp | 0.03% | 0.00 | 0.00 | 0.00 | 0.00 | 0.00 | 0.00 | 0.32 | 0.00 | 0.00 | **0.70** |
| | 0.05% | 0.00 | 0.00 | 0.00 | 0.00 | 0.00 | 0.00 | 0.44 | 0.00 | 0.00 | **0.70** |
| | 0.1% | 0.00 | 0.00 | 0.00 | 0.00 | 0.00 | 0.00 | 0.63 | 0.00 | 0.00 | **0.70** |
| | 0.5% | 0.00 | 0.00 | 0.04 | 0.00 | 0.00 | 0.00 | 0.61 | 0.08 | 0.00 | **0.71** |
| ML-10M | 0.03% | 0.00 | 0.00 | 0.00 | 0.00 | 0.00 | 0.00 | 0.00 | 0.00 | 0.00 | **0.40** |
| | 0.05% | 0.00 | 0.00 | 0.00 | 0.00 | 0.00 | 0.00 | 0.00 | 0.00 | 0.00 | **0.99** |
| | 0.1% | 0.00 | 0.00 | 0.00 | 0.00 | 0.00 | 0.00 | 0.00 | 0.00 | 0.00 | **0.99** |
| | 0.5% | 0.00 | 0.00 | 0.00 | 0.00 | 0.00 | 0.00 | 0.00 | 0.00 | 0.00 | **0.99** |
| ML-20M | 0.03% | 0.00 | 0.00 | 0.00 | 0.00 | 0.00 | 0.00 | 0.00 | 0.00 | 0.00 | **0.99** |
| | 0.05% | 0.00 | 0.00 | 0.00 | 0.00 | 0.00 | 0.00 | 0.00 | 0.00 | 0.00 | **0.99** |
| | 0.1% | 0.00 | 0.00 | 0.00 | 0.00 | 0.00 | 0.00 | 0.00 | 0.00 | 0.00 | **0.99** |
| | 0.5% | 0.00 | 0.00 | 0.00 | 0.00 | 0.00 | 0.00 | 0.00 | 0.00 | 0.00 | **0.99** |

**(e) HiCS**

| Dataset | Attack size | None | Random | Popular | Bandwagon | RAPU-G | RAPU-R | FedRec-Attack | Pip-Attack | PSMU | Poison-FRS |
|---|---|---|---|---|---|---|---|---|---|---|---|
| Steam | 0.03% | 0.00 | 0.00 | 0.00 | 0.00 | 0.00 | 0.00 | 0.00 | 0.00 | 0.00 | **0.98** |
| | 0.05% | 0.00 | 0.00 | 0.00 | 0.00 | 0.00 | 0.00 | 0.00 | 0.00 | 0.00 | **0.98** |
| | 0.1% | 0.00 | 0.00 | 0.00 | 0.00 | 0.00 | 0.00 | 0.28 | 0.00 | 0.00 | **0.99** |
| | 0.5% | 0.00 | 0.00 | 0.03 | 0.00 | 0.00 | 0.00 | 0.93 | 0.01 | 0.00 | **0.99** |
| Yelp | 0.03% | 0.00 | 0.00 | 0.00 | 0.00 | 0.00 | 0.00 | 0.19 | 0.00 | 0.00 | **0.70** |
| | 0.05% | 0.00 | 0.00 | 0.00 | 0.00 | 0.00 | 0.00 | 0.53 | 0.00 | 0.00 | **0.74** |
| | 0.1% | 0.00 | 0.00 | 0.00 | 0.00 | 0.00 | 0.00 | 0.57 | 0.00 | 0.00 | **0.75** |
| | 0.5% | 0.00 | 0.00 | 0.02 | 0.00 | 0.00 | 0.00 | 0.55 | 0.16 | 0.00 | **0.76** |
| ML-10M | 0.03% | 0.00 | 0.00 | 0.00 | 0.00 | 0.00 | 0.00 | 0.00 | 0.00 | 0.00 | **0.38** |
| | 0.05% | 0.00 | 0.00 | 0.00 | 0.00 | 0.00 | 0.00 | 0.00 | 0.00 | 0.00 | **0.99** |
| | 0.1% | 0.00 | 0.00 | 0.00 | 0.00 | 0.00 | 0.00 | 0.00 | 0.00 | 0.00 | **0.99** |
| | 0.5% | 0.00 | 0.00 | 0.00 | 0.00 | 0.00 | 0.00 | 0.00 | 0.00 | 0.00 | **0.99** |
| ML-20M | 0.03% | 0.00 | 0.00 | 0.00 | 0.00 | 0.00 | 0.00 | 0.00 | 0.00 | 0.00 | **0.99** |
| | 0.05% | 0.00 | 0.00 | 0.00 | 0.00 | 0.00 | 0.00 | 0.00 | 0.00 | 0.00 | **0.99** |
| | 0.1% | 0.00 | 0.00 | 0.00 | 0.00 | 0.00 | 0.00 | 0.00 | 0.00 | 0.00 | **0.99** |
| | 0.5% | 0.00 | 0.00 | 0.00 | 0.00 | 0.00 | 0.00 | 0.00 | 0.00 | 0.00 | **0.99** |

**Table 7: Attacking effect evaluated by different metrics on Yelp with FedAvg.**

| Metric | Attack size | None | Rand-om | Popu-lar | Band-wagon | RAPU-G | RAPU-R | FedRec-Attack | Pip-Attack | PSMU | Poison-FRS |
|---|---|---|---|---|---|---|---|---|---|---|---|
| HR@5 | 1% | 0.00 | 0.00 | 0.06 | 0.00 | 0.03 | 0.00 | 0.59 | 0.18 | 0.00 | **0.75** |
|  | 5% | 0.00 | 0.01 | 0.18 | 0.43 | 0.34 | 0.07 | 0.48 | 0.16 | 0.00 | **0.80** |
|  | 10% | 0.00 | 0.01 | 0.23 | 0.62 | 0.44 | 0.18 | 0.56 | 0.24 | 0.00 | **0.81** |
| HR@10 | 0.03% | 0.00 | 0.00 | 0.00 | 0.00 | 0.00 | 0.00 | 0.21 | 0.00 | 0.00 | **0.73** |
|  | 0.05% | 0.00 | 0.00 | 0.00 | 0.00 | 0.00 | 0.00 | 0.23 | 0.00 | 0.00 | **0.73** |
|  | 0.1% | 0.00 | 0.00 | 0.00 | 0.00 | 0.00 | 0.00 | 0.26 | 0.00 | 0.00 | **0.76** |
|  | 0.5% | 0.00 | 0.00 | 0.02 | 0.00 | 0.00 | 0.00 | 0.32 | 0.12 | 0.00 | **0.77** |
| HR@50 | 0.03% | 0.00 | 0.00 | 0.00 | 0.00 | 0.00 | 0.00 | 0.27 | 0.00 | 0.00 | **0.73** |
|  | 0.05% | 0.00 | 0.00 | 0.00 | 0.00 | 0.00 | 0.00 | 0.26 | 0.00 | 0.00 | **0.74** |
|  | 0.1% | 0.00 | 0.00 | 0.00 | 0.00 | 0.00 | 0.00 | 0.30 | 0.01 | 0.00 | **0.76** |
|  | 0.5% | 0.00 | 0.00 | 0.04 | 0.00 | 0.01 | 0.01 | 0.34 | 0.12 | 0.00 | **0.78** |
| NDCG | 0.03% | 0.00 | 0.00 | 0.00 | 0.00 | 0.00 | 0.00 | 0.18 | 0.00 | 0.00 | **0.72** |
|  | 0.05% | 0.00 | 0.00 | 0.00 | 0.00 | 0.00 | 0.00 | 0.20 | 0.00 | 0.00 | **0.74** |
|  | 0.1% | 0.00 | 0.00 | 0.00 | 0.00 | 0.00 | 0.00 | 0.22 | 0.00 | 0.00 | **0.76** |
|  | 0.5% | 0.00 | 0.00 | 0.02 | 0.00 | 0.00 | 0.00 | 0.31 | 0.11 | 0.00 | **0.77** |

