# OpenReview forum: "Poisoning Federated Recommender Systems with Fake Users"
_ACM.org/TheWebConf/2024/Conference — TheWebConf24_

### Official Review · Reviewer_U4F3 · 2023-11-14

**Novelty:** 6
**Technical Quality:** 4

**Review:**

The authors develop an attack in federated recommendation systems that requires no extra information apart from item embeddings obtained from the server. Their fake user based poisoning attack named PoisonFRS promotes the attacker-chosen targeted item in federated recommender systems without requiring knowledge about user-item rating data, user attributes, or the aggregation rule used by the server.
They argue that the requirement for information from genuine users or access to local training data, which can pose significant challenges, especially for recently registered fake users.

In general the paper is well written and procedures, baselines and aggregation strategies well motivated. I like their ablation study on amplification factor, number of popular items, attack initiation times and filler items.

I think the work has some evaluation issues with respect to their chosen baselines. In table 3, most of the baselines perform with HR@5 of 0. This information on all those baselines gives no benefit for the reader except that no baseline seems to work. The authors should consider finding baselines that can provide an actual comparison and show where the performance of the proposed model lies. Alternatively, different evaluation metrics in addition to HR@5 could be shown to check at which point the baselines start or stop working. The reader has no possibility to tell whether there is an issue with the implementation, the assumptions made, or other things that prevent the baselines to work at all. In addition, theres little information about the actual fake users that should be added. From Figure 4, it seems like there are only 10 fake users in the whole approach.

**Questions:**

Will their fake user data and code be made available?

The authors mention:
''We can observe from Table 3 and Table 6 that centralized recommender system-based attacks almost show no attacking effect, which means attacks tailored to FedRecs are quite needed.''
and
''From Table 3 and Table 6, we conclude that for most baseline attacks, their effect will be weakened to some extent when facing defensive aggregation rules''

However I do not think the conclusion can be easily understood for the reader. Maybe the authors can expand on how they develop those conclusions.

Can figure captions be edited to contain more information, such that they can be read in isolation from the text? I think this would be beneficial for the reader.

**Reviewer Confidence:**

2: The reviewer is willing to defend the evaluation, but it is likely that the reviewer did not understand parts of the paper

**Scope:**

4: The work is relevant to the Web and to the track, and is of broad interest to the community

---

### Official Review · Reviewer_kxR6 · 2023-11-21

**Novelty:** 5
**Technical Quality:** 5

**Review:**

This paper focuses on the security issue of Federated recommender system and proposes a new poisoning attack method named PoisonFRS. Instead of requiring additional information related to genuine users’ training data or genuine users’ training data or the distribution of items like recent studies, the attacker can only insert fake users into the recommender system and fetch the item embedding from the server. To mitigate the need for fake users to have local data, PoisonFRS proposes to generate a target item embedding and computes the model updates directly aimed at aligning the embedding of the target item with the target embedding. To evade detection, PoisonFRS also selects filler items and update their embedding updates aiming at maintaining the original embeddings of other items.

pros:

1. This paper conducts adequate experiments on four real-world datasets to evaluate the effectiveness of the proposed PoisonFRS. In addition, the authors compare the proposed PoisonFRS with five traditional poisoning attack and three state-of-the-art poisoning attacks on FedRecs using different robust aggregation rules.
2. The proposed PoisonFRS requires only the item embeddings, instead of requiring any additional information related to genuine users, which is more realistic compared with recent studies.

cons:

1. The attack reults demonstrate the proposed PoisonFRS is an effective poisoning attack method, yet the adaptive defense against this attack is not discussed in this paper.
2. I still wonder why this method works. The kernel of this method lies on the assumption that the average of all item features are unpopular and popular items must exhibit significant dissimilarity from the average item feature. However, there is no proof of the authenticity of this assumption in this paper. Moreover, the parameter $\lambda$ is also a bit confusing.

**Questions:**

1. Adaptive defense against the proposed PoisonFRS requires more discussion or experimentation.
2. More experiments are needed verify the difference between the embeddings of popular projects and the average embedding in the real world. A visualization of item embeddings with varying popularity is sufficient.
3. The attack performamce inproves as the parameter $\lambda$ improves. I am wondering whether the large $\lambda$ would make the value in the target embedding is not in the same level as the size of other embeddings. More explanation about it would be better.

**Reviewer Confidence:**

3: The reviewer is confident but not certain that the evaluation is correct

**Scope:**

4: The work is relevant to the Web and to the track, and is of broad interest to the community

---

### Official Review · Reviewer_SNHk · 2023-11-23

**Novelty:** 5
**Technical Quality:** 5

**Review:**

To develop an attack that requires no extra information apart from item embeddings obtained from the server, this paper introduces a novel fake user based poisoning attack named PoisonFRS to promote the attacker-chosen targeted item in federated recommender systems without requiring knowledge about user-item rating data, user attributes, or the aggregation rule used by the server. Extensive experiments on multiple real-world datasets demonstrate that PoisonFRS can effectively promote the attacker-chosen targeted item to a large portion of genuine users and outperform current benchmarks that rely on additional information about the system.
Pros:
1. Authors introduce a novel poisoning attack on FedRecs that uses fake users, requiring no prior knowledge of genuine user information or access to local training data.
2. Authors systematically evaluate the performance of our proposed attack under various settings, and authors find that PoisonFRS significantly outperforms baseline attacks.
3. Extensive experiments demonstrate that authors proposed PoisonFRS could promote the targeted item to a large fraction of genuine users with a small proportion of fake users, and authors attack cannot be detected by the server.
Cons:
1. Although this article considers many scenarios to verify the effectiveness of PoisonFRS, the text does not consider the issue of new users. How do authors distinguish between new users and fake users? Please provide more detailed explanations from the authors. Can the PoisonFRS algorithm still operate normally when encountering new users.
2. In Section 4, I suggest that the author introduce a specific or intuitive example to simplify the understanding of the motivation behind the algorithm and its specific steps proposed in this article.

**Questions:**

1. As is well known, there is very little information about new users. How does the author define false users to prevent new users from being mistakenly harmed as false users?
2. Why is the number of fillers set to 59 in authors' proposed poison FRS and all baseline attacks?

**Reviewer Confidence:**

4: The reviewer is certain that the evaluation is correct and very familiar with the relevant literature

**Scope:**

4: The work is relevant to the Web and to the track, and is of broad interest to the community

---

### Official Review · Reviewer_sUXq · 2023-11-23

**Novelty:** 5
**Technical Quality:** 5

**Review:**

The paper proposes a poisoning attack on federated recommender systems that allows to promote an item chosen by an attacker to genuine users. The attack uses a small amount of fake users and carefully crafted updates to the local models to influence the recommendations issued by the global model.

The paper is well written and structured but could benefit from improved explanations of some important concepts.

The contributions are clearly stated and the author's introduce preliminary technical details on the concepts they use. However, I believe that some theoretical aspects should be explained further/better: for instance, subsection 2.1 presents federated recommender systems but lacks depth and explanations. I believe this subsection is important as it lays the foundation for a better understanding of the paper. In 4.2, key terminologies should be further described: for example, when the authors talk about "item features", what do the features include? In Section 2, the authors denote $v_i$ to represent the embeddings of item $i$ but in 4.2.1 and 3.2.3, $v^l_i$ and $v_{t}^{'}$ are the "global model at the $l$-th global round" and the "target model" respectively.

The presented methodology looks robust and the authors compare their algorithm to multiple existing works in the literature along with multiple aggregation methods. In all the cases presented, the authors' technique outperforms existing works by a significant margin. The author's choices are well warranted and shown to work through experimentation on real-world datasets. Furthermore, the authors validate the effect of multiple hyper parameters, showing the robustness of their work.
In my opinion, the major drawback here is that the baseline's choices are not justified well enough. I also found the "filler items" paragraph to be complicated to understand, despite the concept being straightforward. Finally, in 4.1, the authors assume that most popular items share roughly similar features: I believe that this claim should be verified or proven in the results section.

I also think the paper would benefit from a small subsection that addresses the limitations of the author's technique and potentially present some suggested defenses.

**Questions:**

a. In my opinion, 4.2.2 and 4.2.3 need reworking. In 4.2.2, I'm not sure I understand what is minimized. What exactly does $v_i$ represent? I feel like a clear definition of what $v$ stands for would clarify multiple aspects of this work.
b. What are the reasons for chosing the mentioned baseline attacks? What is their prevalence in the SOTA's benchmarks? Are they commonly used as a comparing ground?
c. Table 4 shows that the baselines have a significantly different top-5 hit ratio on different datasets, while the author's technique remains relatively stable. What is the reason for this? More precisely, FedRecAttack reaches a score of 0.91 on the Steam dataset but stagnates around 0 and 0.32 at maximum on other datasets. PoisonFRS on the other hand remains stable between 0.72 and 1.00. Is there a reason for this? What were the evaluation conditions, for example, of FedRecAttack? Are they similar to the paper's threat model?
c. Why are the "Median" and "FedAvg" aggregation techniques chosen for the results in Section 5.2? Why not the other mentioned aggregation algorithms?

**Reviewer Confidence:**

2: The reviewer is willing to defend the evaluation, but it is likely that the reviewer did not understand parts of the paper

**Scope:**

4: The work is relevant to the Web and to the track, and is of broad interest to the community

---

### Official Review · Reviewer_Ggd6 · 2023-11-23

**Novelty:** 5
**Technical Quality:** 5

**Review:**

Summary: This paper proposes a novel attack for federated recommender system that promotes items of interest by fake users. Specifically, the proposed method create user-side model updates that could drag the recommender model to promote items of interest. Experiment on Steam, Yelp, and ML-10/20M demonstrate the proposed method is suprior to various baselines, and can break current defenses for recommender systems.

Strength:
- [S1] The stting of the attack model (i.e. creating only user-side updates) is realistic and interesting.
- [S2] The paper is paired with a wide selection of baselines on representative datasets to back the claim well.

Weakness:
- [W1] The claim that the proposed method can remain undetected based on T-SNE visualization seems a bit weak.

**Questions:**

See W1 - is there any good justification on using T-SNE as the detection baseline?

**Ethics Review Description:**

no cerns

**Reviewer Confidence:**

2: The reviewer is willing to defend the evaluation, but it is likely that the reviewer did not understand parts of the paper

**Scope:**

4: The work is relevant to the Web and to the track, and is of broad interest to the community

---

### Decision · Program_Chairs · 2024-01-22

**Decision:**

Accept

**Comment:**

This paper delves into the issue of targeted poisoning attacks on federated recommendation systems, a topic that has garnered significant attention in recent times. Notably, the paper introduces an innovative attack strategy termed PoisonFRS, designed to manipulate federated recommender systems by leveraging item embeddings obtained solely from the server, thereby eliminating the need for additional information. Through extensive experiments conducted on multiple real-world datasets, the paper compares PoisonFRS with existing targeted poisoning attack methods, demonstrating its promising results and showcasing its advantages.


 However, certain aspects of the paper could benefit from further refinement. Firstly, the absence of a discussion or proposal for a new countermeasure is notable, especially given that the experimental results indicate the ineffectiveness of existing defense methods against PoisonFRS. Addressing this gap would contribute to the completeness of the paper by considering potential defenses to mitigate the impact of the introduced attack. Furthermore, the motivation and rationale behind the main algorithm in Section 4 should be more explicitly illustrated and clarified to enhance clarity and comprehension. A more detailed explanation of the underlying principles would aid readers in grasping the intricacies of the proposed approach. Additionally, it would be beneficial for the paper to discuss the limitations of the PoisonFRS attack method. For instance, highlighting scenarios where the method may not be applicable, such as those involving new users, would provide a more comprehensive understanding of its scope and applicability.